# Dice-GAN: Generative Adversarial Network with Diversity Injection and Consistency Enhancement

## Abstract

In the field of natural language description tasks, one challenge for text-to-image modeling is to generate images that are both of high quality and diversity and maintain a high degree of semantic consistency with the textual description. Although significant progress has been made in existing research, there is still potential for improving image quality and diversity. In this study, we propose an efficient attention-based text-to-image synthesis model based on generative adversarial network named Dice-GAN. To improve the diversity of image generation, we design a diversity injection module, which injects noise several times during the image generation process, fuses the noise with the textual information, and incorporates a self-attention mechanism to help the generator maintain global structural consistency while enhancing the diversity of the generated image. To improve the semantic consistency, we designed a consistency enhancement module, which enhances the semantic consistency of image generation by combining word vectors and a hybrid attention mechanism to achieve dynamic weight adjustment for different image regions. We conducted experiments on two widely used benchmark datasets, CUB and COCO. Dice-GAN demonstrated significant superiority in improving the fidelity and diversity of image generation compared to the existing approaches.

## 1 Introduction

Generating images from textual descriptions in natural language is a challenging cross-modal generative task. In recent years, with the advancement of Generative Adversarial Network (GAN) technology (Goodfellow et al., 2014), the field has been significantly developed (Xu et al., 2018; Li et al., 2019b; Liang et al., 2020; Jiang et al., 2024). Furthermore, the outcomes of studies focusing on text-driven image generation have found wide-ranging applications across various domains, encompassing diverse image synthesis tasks such as image manipulation (Liu et al., 2020; Zhang et al., 2020a), facial synthesis (Karras et al., 2018; Zhang et al., 2020b), image restoration (Denton et al., 2016; Yu et al., 2019a), and image enhancement (Zhang et al., 2017b; 2018a;b). As a result, this research area has become one of the most active research topics in the past few years.

Nonetheless, text-to-image generation confronts two principal challenges:

First, the text-to-image synthesis process involves a complex one-to-many mapping relationship. As textual descriptions typically cover only a fraction of an image's features, generating images introduces significant uncertainty. This uncertainty naturally leads to varied image outputs. However, existing models (Hinz et al., 2020; Qi et al., 2021) often introduce noise at the network's inception to exploit this diversity. Unfortunately, the effectiveness of noise diminishes over the course of training, potentially compromising image diversity. A notable consequence of this challenge is the repetitive generation of identical or highly similar images when provided with the same textual input, as illustrated in Fig. 1. This example highlights instances of limited diversity. The recurring issue of producing similar images from identical textual prompts underscores the limitations of current approaches in maintaining diversity throughout image generation.

Second, maintaining the visual quality and semantic consistency of images remains a major challenge when it comes to the generation of complex scenes. Current approaches (Cheng et al., 2020;

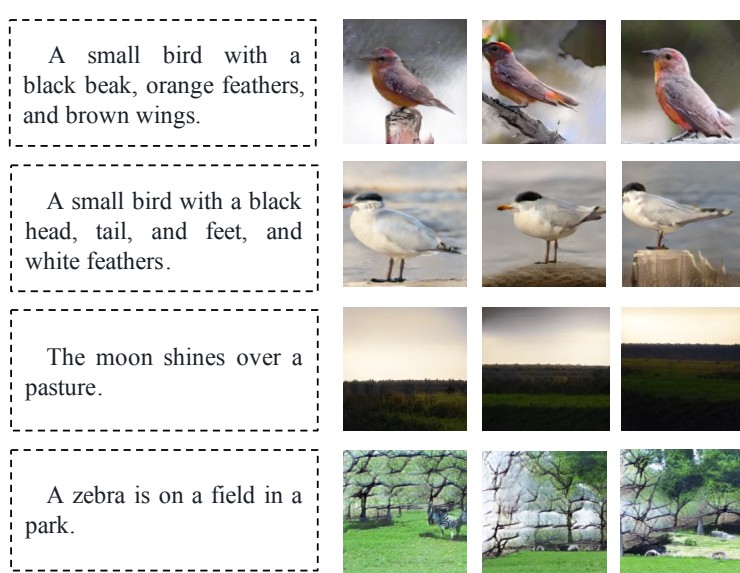

Figure 1: Example of an image generated by a model lacking diversity

Li et al., 2019a; Xu et al., 2018; Zhu et al., 2019) commonly employ cross-modal attention mechanisms aimed at guiding the model to synthesize the corresponding fine-grained details based on the features of the input sentences. However, most of the attention mechanisms in these approaches focus primarily on the spatial dimension and tend to ignore the interrelationships between feature channels. Since each feature channel is directly related to the final image modality, neglecting a reasonable ordering of the importance of individual channels may lead to confusion in the generated image, which in turn leads to a degradation of semantic consistency.

To address with the above problems, this paper proposes a novel generative adversarial network (Dice-GAN). To alleviate the problem of degradation of image diversity, we introduce a diversity injection (DI) module into the generator. This module injects both noise broadcasts and sentence vectors into the generator to balance the effect of noise on the visual process. In addition, the modules incorporate a self-attention mechanism to enhance the structural rationality of the layout of the synthesized image, thus enhancing the diversity of the image generation. To improve the visual quality and semantic consistency of image generation, we design a consistency enhancement (CE) module. This module allows the model to dynamically adjust the weights of different image regions according to the semantic information of the input text by fusing word vectors and hybrid attention mechanisms, thus rationalizing the interplay between primary and secondary visual features, so that the generated images can better achieve semantic consistency while ensuring visual quality. Ultimately, extensive experiments on two datasets show that Dice-GAN can generate images with high diversity, high-quality visual performance, and enhanced semantic consistency aligned with user-provided textual descriptions.

The main contributions of this study can be summarized in the following three aspects:

1. We design a diversity injection module. This module injects noise and sentence vectors into the generator multiple times during the image generation process and incorporates a self-attention mechanism to balance the effect of noise, thus enhancing the diversity of image generation.

2. We propose a consistency enhancement module. This module orchestrates the intricate interplay between primary and secondary visual features by dynamically adjusting the weights assigned to different image regions, thus improving semantic consistency while ensuring the visual quality of image generation.

3. We conducted extensive experimentation on two widely used benchmark datasets. The outcomes demonstrate that Dice-GAN surpasses existing approaches in terms of performance.

This validation underscores the effectiveness and progressiveness of the proposed approach in the field of text-to-image synthesis.

## 2 RELATED WORK

Within the contemporary realm of image generation, text-to-image generation techniques can be categorized into two main groups based on the complexity and structural intricacies involved in the generation process: single-stage generation models and multi-stage generation models. The single-stage generation model directly crafts an image from a provided textual description by leveraging a generator and a discriminator. This approach is characterized by its simplicity and directness in image generation. In contrast, the multi-stage generation model adopts a more hierarchical approach to generation. This model comprises several pairs of generators and discriminators, each allocated to a distinct phase in the image creation process. Typically executed sequentially from coarse to intricate details, each stage iteratively refines and enhances the image based on the preceding stage, culminating in the synthesis of a high-fidelity image.

In single-stage generative modeling, GAN-INT-CLS (Reed et al., 2016) achieves the first single-stage text-to-image synthesis task for training conditional GANs by combining an image-text matching discriminator and text streaming interpolation learning, followed by DCGAN (Radford et al., 2016) which demonstrates its strong potential in unsupervised learning by combining deep convolutional networks. An auxiliary classification loss is introduced in TAC-GAN (Dash et al., 2017) to enhance the image quality. HDGAN (Zhang et al., 2018b) innovatively introduces a hierarchically nested discriminator in the network structure to assist the generator in training and capturing complex image semantic information. DF-GAN (Tao et al., 2022) employs a target-aware discriminator to enhance the semantic consistency between text and image. DE-GAN (Jiang et al., 2024) employs a conditional channel attention module to integrate the textual and visual information to make the final generated image more visually logical and thus better semantically aligned with the given text.

Among the multistage generative models, StackGAN (Zhang et al., 2017b) and its improved Stack-GAN++ (Zhang et al., 2018a) use a two-stage generative process to achieve high-quality image synthesis. StyleGAN (Karras et al., 2019) enhances the model's ability to control the diversity of the generated images by introducing a style blending mechanism and Adaptive Instance Normalization (AdaIN) technique, which enables more flexible production of images with rich variations. Mani-GAN (Li et al., 2020) introduces the affine combination module and detail correction module in the multistage model structure to generate images with higher semantic consistency. LAPGAN (Denton et al., 2015) utilizes cascaded convolutional networks in the framework of the Laplace pyramid to generate images in a coarse-to-fine manner, with each layer using GAN to train a separate generative convolutional network model. RiFeGAN (Cheng et al., 2020) enriches the semantic descriptions by embedding a priori knowledge and combines it with a caption-matching approach to achieve the generation of images with a high degree of matching to the descriptions. DAE-GAN (Ruan et al., 2021) efficiently solves the problem by integrating a global refinement module and an aspect-aware local refinement module. The problem is that aspect information may be ignored during text-to-image synthesis.

In the field of image processing, the attention mechanism has become a key technique to improve the representation ability of neural networks (Chen et al., 2018; Hu et al., 2018; Wang et al., 2020; Woo et al., 2018). It has been applied in several subfields, including image translation (Ma et al., 2020; Yang et al., 2020; Yang & Qi, 2021; Emami et al., 2020), image caption generation (Shrimal & Chakraborty, 2021; Yang et al., 2021b) and visual questioning (Gao et al., 2019; Lee et al., 2021; Yang et al., 2021a; 2019; Yu et al., 2019b). Especially in text-to-image generation tasks, cross-modal attention mechanisms play an important role in improving the visual quality of images and ensuring the semantic consistency between the generated images and the given text descriptions. AttnGAN (Xu et al., 2018) achieves conditional generation at the word level by introducing word-level attention to help the generator produce images with a stronger relevance to the given text. SAGAN (Zhang et al., 2019a) introduces a self-attention generative adversarial network, which generates high-quality images by applying remote dependency modeling to the image generation task in an attention-driven manner. Unlike the above models, ControlGAN (Li et al., 2019a) is able to decouple different visual attributes by introducing a word-level spatial and channel-attention mechanism-driven generator and enables the model to focus on generating image subregions corre-

sponding to specific textual descriptions. DR-GAN (Tan et al., 2022), on the other hand, combines a spatial self-attention mechanism in a semantic disentanglement module to help the generator extract the critical information required for image generation. Despite the performance improvement of these attention mechanism-based models, they still have a large number of hidden states during the training process, which limits the further optimization of the models.

Compared to the existing approaches, our proposed Dice-GAN utilizes a single-stage model structure to demonstrate higher visual quality and better diversity in image synthesis. Our proposed consistency enhancement module can effectively increase the weights of text features and enhance the efficiency of model training, thus achieving performance optimization.

## 3 DICE-GAN

We propose a Dice-GAN model for text-to-image synthesis. The Dice-GAN model's architecture incorporates a diversity injection (DI) module and a consistency enhancement (CE) module, essential components for the model's comprehensive functionality. The overall architecture of the Dice-GAN model proposed in this study is depicted in Fig. 2.

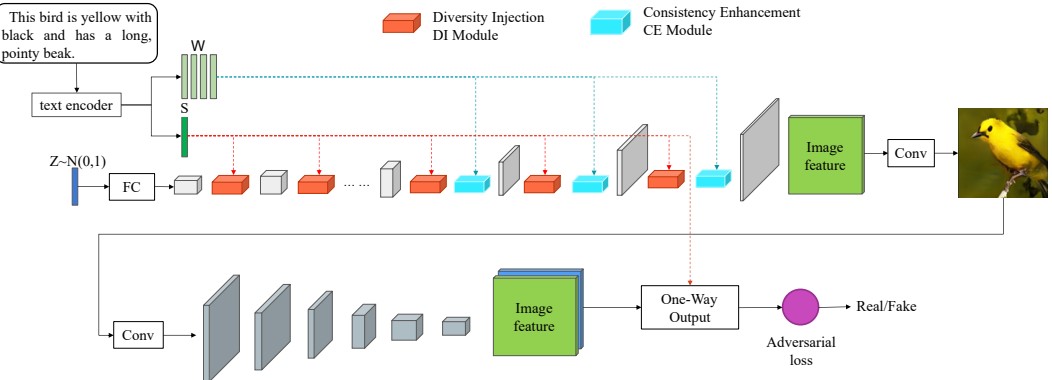

Figure 2: Overall architecture of Dice-GAN model

Initially, we extract features from the input textual descriptions, deriving sentence-level feature vectors denoted as $S$ and word-level feature vectors denoted as $W$, as outlined in previous works (Xu et al., 2018; Reed et al., 2016). Subsequently, we generate latent vectors from a standard normal distribution $z$, which serve as inputs to the Dice-GAN model.

### 3.1 DIVERSITY INJECTION MODULE

During the image generation process, we feed the sentence vector $S$ into the Diversity Injection (DI) module. Here, the textual information is amalgamated with visual features and noise via a feature fusion layer and a noise broadcasting mechanism, thereby intensifying the influence of noise on the diversity of image generation. Furthermore, a self-attention mechanism is integrated within this module to harmonize the global layout structure of the image. As a result, Dice-GAN demonstrates significant advantages in generating images with increased diversity when contrasted with previous methodologies. The structure of DI module is shown in Fig. 3.

We introduce a feature fusion layer within the module to acquire the scale parameters $\gamma_c(S)$ and bias parameters $\beta_c(S)$ for the textual information, denoted by $S$, in our approach. Here, the input image features are denoted as $F_c \in \mathbb{R}^{C \times H \times W}$, where $C$ signifies the channel count, $H$ denotes the height, and $W$ represents the width of the image. Upon undergoing processing by the feature fusion layer, the modified image features $F_c'$ are expressed as shown in Equation 1.

$$F_c' = \gamma_c(S) \times F_c + \beta_c(S) \tag{1}$$

Here, the symbol $c$ signifies the feature map $F_c$ corresponding to each channel within the range $(c = 1, 2, \ldots, C)$. By fine-tuning the feature fusion layer, the resultant output feature map $F_c'$ aligns

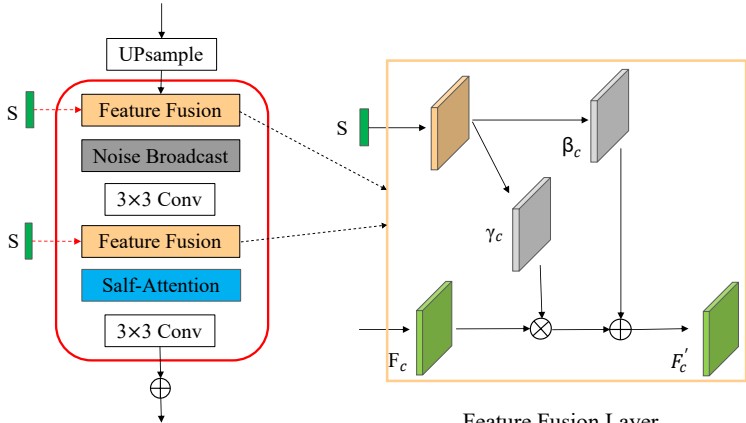

Figure 3: Structure of the diversity injection module

more cohesively with the provided textual information. To guarantee effective feature fusion, we incorporate two feature fusion layers within each DI module.

Following the fusion of features, to uphold diversity in the generation procedure, we introduce noise $N$ into the processed feature map $F_c'$. The injection process of this noise is depicted in Equation 2.

$$I = F_c' + \sigma N \tag{2}$$

where $\sigma$ represents a trainable parameter initialized to 0 at the initial stage of the training process. By incorporating this adjustable parameter $\sigma$, our objective is to control the level of noise injection, thereby preventing potential deterioration in visual quality arising from excessive noise injection throughout the image generation process. Furthermore, we incorporate a self-attention mechanism to capture long-range dependencies within the image during the generation process, thereby enhancing global coherence in the resulting image.

## 3.2 CONSISTENCY ENHANCEMENT MODULE

To enhance the consistent generation of image features and textual information, we consider improving the model from both channel and spatial perspectives. In the Consistency Enhancement (CE) module, we successfully integrate the word vector $W$ into the conditional channel attention mechanism, which is used to identify and enhance the most important feature channels in the generator to improve the quality of the generated images. By learning the importance of each channel, the model can pay more attention to the information that is crucial for image generation while suppressing irrelevant features. This is combined with a spatial attention mechanism to ensure that high-level and low-level features complement each other in generating the image, enhancing the detail and structure of the image. This integration aims to improve the visual quality throughout the image generation process. Fig. 4 provides a visual representation of the integrated structure of the CE module.

In this module, we use the input image features $F_c \in \mathbb{R}^{C \times H \times W}$ and word vectors $W = \{w_{1,j} w_2, \ldots, w_N\}$, in which, each word vector $w_i \in \mathbb{R}^D$, where $D$ and $R$ denote the dimension and number of word vectors, respectively. The detailed computational flow of the CE module is as follows:

**Word vector embedding stage.** First, the set of word vectors $W$ is processed through the fully-connected layer to produce the aggregation matrix $W_{agg}$.

**Hybrid attention feature generation stage.** We merge the conditional channel attention mechanism with the spatial attention mechanism to enable the model to dynamically emphasize crucial channels and spatial positions within the input features. The specific procedure involves transposing the aggregation matrix $W_{\text{agg}}$ to yield $W_{\text{agg}}^T$, followed by comparing it with the maximum pooling outcome $F_{\text{max}}^c$ and the average pooling outcome $F_{\text{avg}}^c$ of the input features $F_c$. Matrix multiplication is executed to derive two feature aggregation matrices, namely $G_{\text{max}}^c$ and $G_{\text{avg}}^c$. The maximum

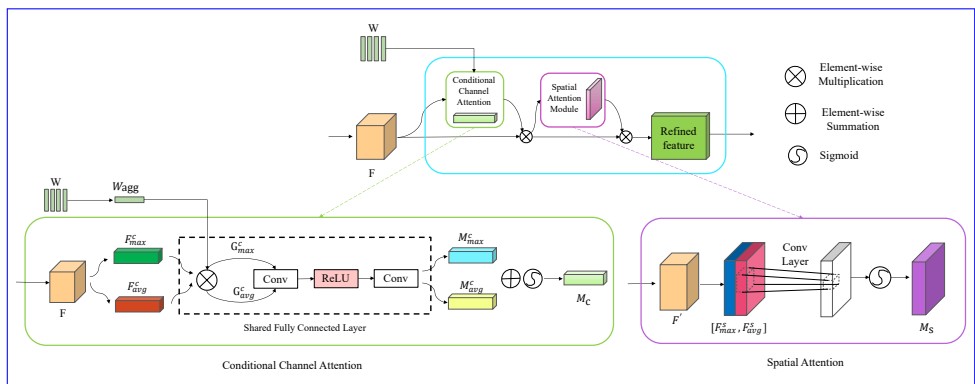

Figure 4: Structure of the consistency enhancement module

pooling operation retains the maximum values for each channel, which represent the most salient features in the feature map, such as critical parts of the image or edge information. The average pooling operation calculates the average of all values for each channel, which reflects the overall characteristics of the feature map. Average pooling captures the global information in the feature map, including background and texture. Thus, it can preserve the background information in the feature map and help the model better understand the overall structure.

Subsequently, weight coefficients are assigned to each channel to formulate two channel attention maps, denoted as $M_c^{\max}$ and $M_c^{\text{avg}}$. These weight coefficients $g_*^{i,j}$ are determined jointly by the $i$-th channel of $F$ and the $j$-th word of $W$. Through an element-wise summation operation, $M_c^{\max}$ and $M_c^{\text{avg}}$ are combined, and the final conditional channel attention map $M_c$ is obtained by applying the sigmoid function $\sigma$. The computational steps are outlined in Equation 3 and Equation 4.

$$M_c^* = \{m_1^*, m_2^*, ..., m_C^*\}, m_i^* = \sum_{j=1}^{T} g_*^{i,j} \tag{3}$$

$$M_c = \sigma(M_c^{\max} + M_c^{avg}) \tag{4}$$

Then, the conditional channel attention map $M_c$ is multiplied with the original input features $F$ to generate a new feature set $F'$, which serves as input to the spatial attention module. To delve deeper into spatial information extraction, we compress the channels of $F'$ and conduct average pooling ($F_{\max}^s$) and maximum pooling ($F_{\text{avg}}^s$) operations to obtain two compressed feature maps. These maps are concatenated along the channel dimensions and convolved with a $3 \times 3$ convolution kernel to condense the feature dimensions. Subsequently, a sigmoid function is applied to the output to obtain the spatial attention map $M_s$.

Finally, $M_s$ is multiplied with $F'$ to yield the ultimate image features adjusted by spatial attention, denoted as $F''$. This signifies the conclusive image features refined by spatial attention. The computational steps are delineated in Equation 5 and Equation 6.

$$M_s = \sigma(f^{3\times3}([F_{max}^s, F_{avg}^s])) \tag{5}$$

$$F'' = M_s \times F' \tag{6}$$

where $f^{3\times3}$ denotes the $3 \times 3$ size of the convolution kernel. The image features refined by spatial attention, denoted as $F''$, are forwarded to the subsequent module for image generation process.

Through this series of operations, the CE module is able to effectively fuse textual descriptions and visual content by dynamically adjusting the weights of image features to retain and emphasize critical information while reasonably reducing the weights of non-critical features. This processing not only improves the visual quality of the generated image, but also enhances the consistency between the image and the textual description.

### 3.3 Objective Function

Drawing from previous studies (Tao et al., 2022), we integrate the Match-Aware Gradient Penalty (MA-GP) and One-Way Output techniques into our model training strategy to enhance our network architecture. The optimization of our network structure is guided by loss functions for the generator (Equation 7) and discriminator (Equation 8), respectively.

$$L_G = -\mathbb{E}_{G(z) \sim \mathbb{P}_g}[D(G(z), S)] \tag{7}$$

$$
\begin{aligned}
L_D = &-\mathbb{E}_{x \sim \mathbb{P}_r}[h_1(D(x, S))] \\
&- \frac{1}{2}\mathbb{E}_{G(z) \sim \mathbb{P}_g}[h_2(D(G(z), S))] - \frac{1}{2}\mathbb{E}_{x \sim \mathbb{P}_{mis}}[h_2(D(x, S))] \\
&+ k\mathbb{E}_{x \sim \mathbb{P}_r}[(\| \nabla_x D(x, S) \| + \| \nabla_S D(x, S) \|)^p]
\end{aligned}
\tag{8}
$$

where $G(z)$ denotes the generator $G$ by extracting potential vectors from the potential space $z$ and mapping it to the image space to generate the image samples, and $\mathbb{P}_g$ is the distribution of the generated image, and $D(\cdot, S)$ is the discriminator network, which receives image and sentence vectors $S$ as input and outputs a match score. The functions $h_1(t) = \min(0, -1 + t)$ and $h_2(t) = \min(0, -1 - t)$ represent the hinge losses, utilized to evaluate the correspondence between the real and generated samples, respectively. Here, $x \sim \mathbb{P}_r$ signifies the real sample, while $x \sim \mathbb{P}_{mis}$ denotes image samples that do not align with the data. The parameters $k$ and $p$ denote the two components of the balanced gradient. The MA-GP incorporates a gradient penalty term into the discriminator's loss function, ensuring stable convergence to the real data distribution when processing matched data. This mechanism safeguards against issues like vanishing or exploding gradients for the generator during training. On the other hand, One-Way Output improves the learning efficiency and convergence speed of the generator.

## 4 Experiments

### 4.1 Experimental setup

**Datasets.** To assess the efficacy of our proposed Dice-GAN model, we conducted experiments using two prominent benchmark datasets: the CUB dataset (Wah et al., 2011) and the MS-COCO dataset (Lin et al., 2014). The CUB dataset comprises 11,788 images representing 200 distinct bird species. These images are segregated into training and testing sets, with the training set comprising 8,855 images and the test set containing 2,933 images. Each bird image is accompanied by ten unique textual descriptions. In contrast, the MS-COCO dataset consists of 80,000 training images and 40,000 test images, each paired with five language descriptions.

**Implementation details.** In our experimental setup, we utilized Ubuntu 20.04 as the operating system, PyTorch 1.11.0 as the deep learning framework, and Cuda 11.3 for GPU acceleration. For hardware, we harnessed the computational power of the NVIDIA RTX 4090 graphics card, boasting 24GB of video memory. For the image generation task, we configured the output image resolution to be $256 \times 256$ pixels to ensure the generated images exhibit adequate clarity. We employed the Adam optimizerr (Kingma & Ba, 2015) to optimize the network parameters, with $\beta_1 = 0.0001$ and $\beta_2 = 0.9$ settings. The learning rate for the generator was set to 0.0001, while for the discriminator, it was set to 0.0004.

**Evaluation metrics.** Building upon prior research (Xu et al., 2018; Zhu et al., 2019), we employ two established evaluation metrics, namely the Inception Score (IS) (Salimans et al., 2016) and the Fréchet Inception Distance (FID) (Heusel et al., 2017) and CLIPScore (Hessel et al., 2021) to assess the performance of our proposed Dice-GAN model. The IS computation leverages the pre-trained Inception v3 network to gauge the visual quality and diversity of generated images. It evaluates the quality and diversity by analyzing the KL divergence between the conditional and marginal distributions of the generated images. A higher IS value signifies that the model's generated images not only possess high visual quality but also demonstrate diversity in category distribution. Conversely, the FID also utilizes the pre-trained Inception v3 network, assessing the authenticity of generated

images and their alignment with provided textual descriptions. This evaluation metric calculates the Fréchet distance between the distribution of synthesized images and the distribution of real images in the feature space. Unlike the IS metric, a lower FID value indicates a closer resemblance between the generated and real images in terms of statistical features, reflecting higher fidelity in the generated images and stronger semantic consistency. CLIPScore is a metric for assessing the consistency between image and text descriptions by calculating the cosine similarity between the CLIP embedding vectors of the generated images and text descriptions. The value of CLIPScore ranges from 0 to 1, where 1 indicates perfect consistency between the image and text descriptions, and 0 means no correlation. Therefore, the higher the CLIPScore value, the higher the consistency between the image and the text description. To fully evaluate the performance of the DI module in improving image diversity, we computed the average LPIPS distance between 3K pairs of images, each generated from the same sentence. Higher LPIPS values indicate greater differences between images, thus reflecting better diversity.

## 4.2 COMPARISON OF EXPERIMENTAL RESULTS

In this research, to provide an objective assessment of the Dice-GAN model's performance in text-to-image synthesis, we conduct a comparative analysis against a selection of cutting-edge text-to-image synthesis approaches. These methods include AttnGAN (Xu et al., 2018), DM-GAN (Zhu et al., 2019), DF-GAN (Tao et al., 2022), DE-GAN (Jiang et al., 2024), StackGAN (Zhang et al., 2017a), StackGAN++ (Zhang et al., 2019b) and StyleGAN (Karras et al., 2019).

Furthermore, we extend our comparative study to diffusion models such as CogView (Ding et al., 2021), DALL-E (Ramesh et al., 2021), and ShiftDDPMs (Zhang et al., 2023), which has also demonstrated notable success in the domain of text-to-image synthesis. Since the training process of CogView and DALL-E is extremely complex and requires a large amount of computing resources, we obtain their pre-trained models from the open source community. All of comparison methodologies have achieved significant success in the field. Through such a comparative analysis, we aim to gain a comprehensive understanding of the performance and relative advantages of Dice-GAN in different models.

### 4.2.1 NUMERICAL COMPARISONS

The comparison results of IS, FID and CLIPScore across different models are shown in Table 1, where the best performance is shown in bold. By comparing with the stacked architectures AttnGAN, StackGAN, StackGAN++, StyleGAN models and the single-stage architectures DM-GAN, DF-GAN, and DE-GAN models on the CUB dataset, our Dice-GAN model demonstrates significant enhancement in the IS metrics from 4.28 to 4.93, the FID metrics from 24.17 to 15.81 and CLIPScore improves from 0.18 to 0.29, demonstrating that Dice-GAN exhibits excellent performance. Converting to the MS-COCO dataset, Dice-GAN excels in FID performance, reducing it from 34.53 to 23.31. However, Dice-GAN slightly lags behind the other methods in terms of IS metrics. This discrepancy can be attributed to an inherent limitation of the IS metric (Zhang et al., 2021): the Inception model for IS computation was pre-trained on the ImageNet dataset, which is typically characterized for a single primary object, in contrast to the combinations of multiple objects that are often found in the MS-COCO dataset. This difference may lead to bias in IS assessment. Notably, our approach produces superior results to the diffusion models CogView, DALL-E, and ShiftDDPMs when compared to these models. In the above study, in addition to evaluating the image synthesis quality of the model, we also examined the operational efficiency of the model, especially the speed of image generation. The results show that the Dice-GAN model not only performs superiorly in terms of image quality and semantic consistency, but also significantly improves the efficiency of image generation. Specifically, compared to other models, Dice-GAN reduces the time for image generation from an average of 23.98 seconds to 9.02 seconds, which means that its image generation speed is improved by about 62.4%. This improvement significantly enhances the efficiency of image generation, making Dice-GAN more efficient and practical in practical applications.

### 4.2.2 ABLATION STUDY

The results of the ablation experiment of DI and CE modules on the CUB dataset are shown in Table 2 and Table 3 respectively.

Table 1: Comparison results of IS, FID, CLIPScore and inference time between the existing models and our model on the CUB and COCO test sets

| Model | CUB | | | MS-COCO | | | Inference Time(s) |
|---|---|---|---|---|---|---|---|
| | IS↑ | FID↓ | CLIPScore↑ | IS↑ | FID↓ | CLIPScore↑ | |
| AttnGAN | 4.28 | 24.17 | 0.18 | 21.16 | 34.53 | 0.19 | 23.98 |
| DM-GAN | 4.51 | 16.83 | 0.21 | **25.38** | 32.64 | 0.20 | 21.63 |
| DF-GAN | 4.62 | 19.40 | 0.23 | 17.96 | 26.79 | 0.24 | 19.70 |
| DE-GAN | 4.88 | 17.52 | 0.22 | 18.33 | 27.41 | 0.21 | 16.52 |
| StackGAN | 3.76 | 50.89 | 0.18 | 8.98 | 33.68 | 0.17 | 22.06 |
| StackGAN++ | 3.82 | 26.30 | 0.18 | 8.46 | 51.62 | 0.18 | 20.24 |
| StyleGAN | 3.89 | 19.36 | 0.24 | 14.85 | 29.09 | 0.26 | 17.32 |
| ShiftDDPMs | 4.42 | 16.09 | 0.25 | 17.74 | 23.85 | 0.25 | 12.64 |
| CogView | 4.62 | 15.95 | 0.25 | 18.03 | 27.23 | 0.24 | 10.21 |
| DALL-E | 4.81 | 15.97 | 0.28 | 17.89 | 27.25 | 0.27 | 9.80 |
| Dice-GAN(ours) | **4.93** | **15.81** | **0.29** | 18.20 | **23.31** | **0.28** | **9.02** |

Table 2: Ablation study on DI module

| Model | IS↑ | FID↓ | LPIPS↑ |
|---|---|---|---|
| Baseline | 4.62 | 19.40 | 0.48 |
| B+1FF | 4.70 | 18.83 | 0.51 |
| B+2FF | 4.79 | 18.52 | 0.55 |
| B+3FF | 4.73 | 18.65 | 0.52 |
| B+SA | 4.76 | 18.49 | 0.54 |
| B+DI | **4.81** | **18.37** | **0.57** |

Table 3: Ablation study on CE module

| Model | IS↑ | FID↓ |
|---|---|---|
| Baseline | 4.62 | 19.40 |
| B+CCA | 4.63 | 19.21 |
| B+SPA | 4.63 | 17.58 |
| B+CE | **4.65** | **16.24** |

Impact of DI: Incorporating the DI module significantly enhances the model's image generation quality, yielding an IS value of 4.81 and an FID value of 18.37. Furthermore, our experiments indicate that an excessive number of feature fusion layers escalates computational load without commensurate benefits. Consequently, we opted for two feature fusion layers to amplify text-information fusion with image features. These refinements bolster the DI module's capacity to produce diverse, high-fidelity images, reinforcing its efficacy and practicality.

Effect of CE: After combining the CE module, the IS value increased from 4.62 to 4.65, and the FID value decreased significantly from 19.40 to 16.24. In the ablation experiments, the addition of either the conditional channel attention mechanism or the spatial attention mechanism alone ignored some of the information to some extent, and the consistency of the model was significantly enhanced after combining the two mechanisms in the CE module for the experiments. In addition, the experiments found that the resolution of the image features generated in the early stage of the model generation is small, the effect of adding the CE module on the consistency enhancement is not obvious, and it will increase the computation time of the model. Therefore, we chose to add the CE module at the stage with a resolution of $64 \times 64$, which can significantly improve the semantic consistency of the model-generated images while maintaining computational efficiency. These findings confirm the effectiveness and usefulness of the proposed CE module in improving the semantic consistency of model-generated images.

### 4.2.3 CASE STUDY

In this case study, we analyzed the CUB and MS-COCO datasets and compared the outputs of AttnGAN, DMGAN, DF-GAN, DE-GAN, StackGAN, StyleGAN, and our proposed Dice-GAN, as shown in Fig. 5. It is found that there are significant differences in the synthesis quality of these models. First, for the CUB dataset, the models have some problems in image generation under the long text of meticulous description. For example, the image in column 1 is missing body parts, the body shape is distorted in the image in column 2, the feather texture is confusing in column 3 and the incongruous body proportions in column 4. The differences in performance between models are still significant when using text with short descriptions, such as inconsistent colors in column 5, while

column 6 results in greater diversity in the generated images due to the lack of specific descriptions of bird types, colors and sizes. In contrast, the Dice-GAN model demonstrates excellent image synthesis ability when processing both long and short text tasks. The model can effectively maintain the integrity and coherence of the subject in the image while generating details with natural gestures and realism, especially when generating bird images. In addition, when processing complex scene synthesis tasks from the MS-COCO dataset, Dice-GAN not only demonstrates excellent semantic consistency under long text descriptions, but can more accurately localize textual features, such as "Milkshake" in column 8 and "Train track" in column 10, but also demonstrates excellent semantic consistency under short text descriptions, such as "Milkshake" in column 8 and "Train track" in column 10. It also ensures the harmony and accuracy of image feature localization in the face of short text descriptions, as demonstrated by "bedroom" in column 11 and "kite" in column 12. This shows that the Dice-GAN model is highly specialized in generating complex scenes with high fidelity.

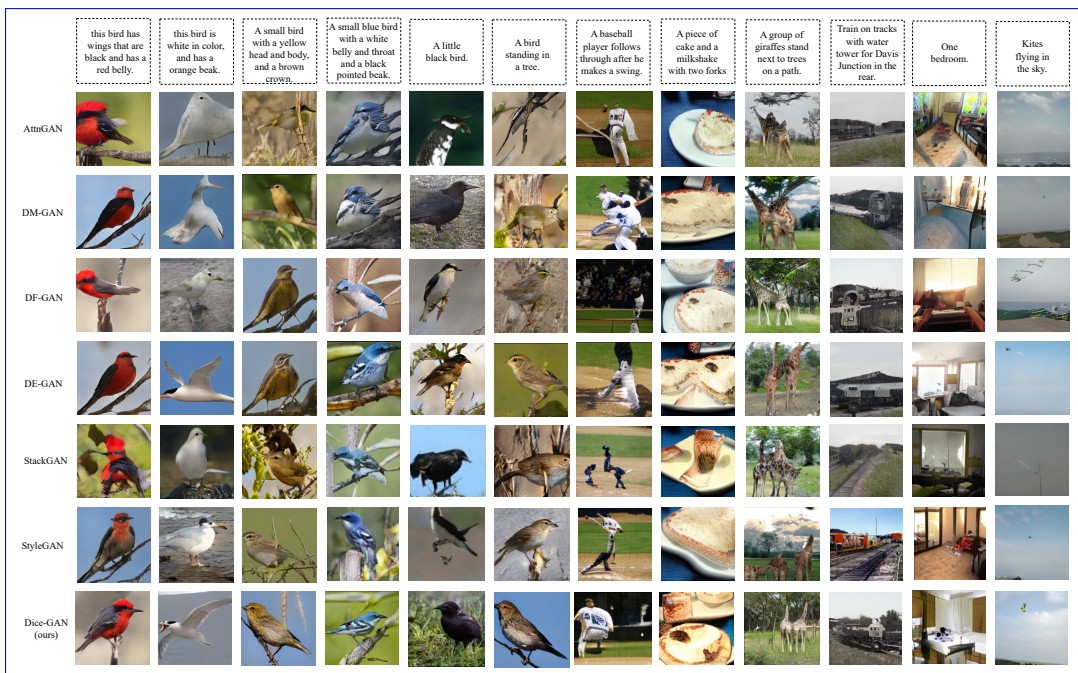

Figure 5: Visualization of images generated by Dice-GAN and state-of-the-art models

## 5 CONCLUSION

This paper introduces Dice-GAN, a novel model tailored for text-to-image synthesis. Operating within a single-stage GAN framework, Dice-GAN directly produces high-resolution images. Central to its architecture is the Diversity Injection (DI) Module, which merges injection noise and textual data within the generator, heightening the stochastic nature of the generated images. Leveraging a self-attention mechanism, the DI module prioritizes global structural coherence, elevating visual quality while preserving image diversity. Furthermore, a Consistency Enhancement (CE) Module is proposed to fine-tune channel weights during image generation. By integrating word vectors into the conditional channel attention mechanism, the CE module bolsters features aligned with textual descriptions. Spatial attention mechanisms are also incorporated to capture inter-regional relationships within images, ensuring spatial consistency and coherence during synthesis, thereby enhancing semantic fidelity. Through experiments on standard datasets, Dice-GAN demonstrates commendable performance in image synthesis, underscoring its efficacy in text-to-image generation tasks.

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

## A APPENDIX

### A.0.1 VISUAL COMPARISON OF THE IMAGENET AND MS-COCO DATASETS

Fig. 6 illustrates a contrast between images found in the ImageNet dataset and the MS-COCO dataset. The Inception model has undergone pre-training on the ImageNet dataset, where images predominantly contain features of a single primary object. This stands in contrast to the MS-COCO dataset, where images frequently depict a combination of multiple objects.

### A.0.2 VISUAL COMPARISON OF ABLATION EXPERIMENTS

To evaluate the performance of the modules proposed in this study in-depth, we performed a visual comparative analysis of the synthesized images. Fig. 7 illustrates the results of this comparison. By comparing the images in rows 1 and 2, we found that, without the DI module, the images generated by the baseline model present monotonicity in terms of background and content. On the contrary, with the introduction of the DI module, the background and pose diversity of the image generation is significantly improved, which indicates that the DI module has a positive effect in enhancing the diversity of the image generation. In addition, the comparison of the images in rows 1 and 3 shows that the application of the CE module improves the semantic consistency of the synthesized

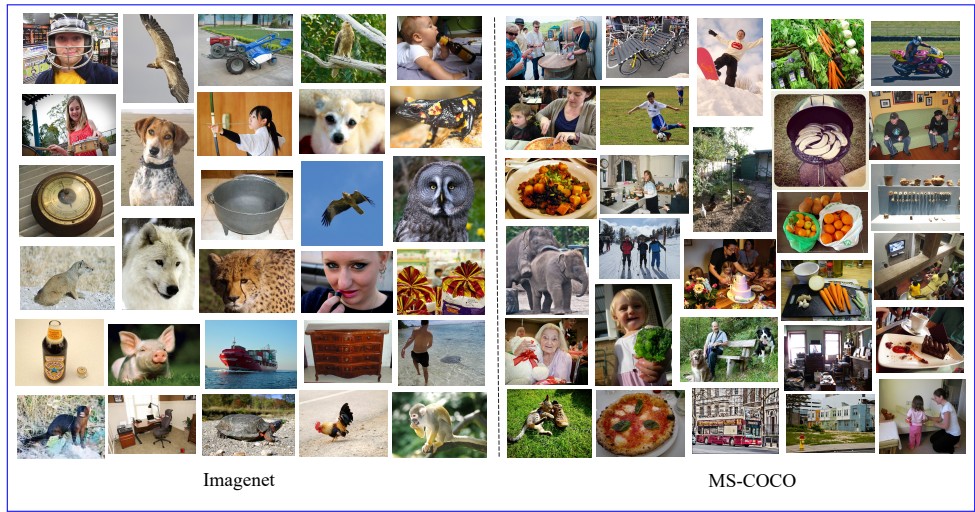

Figure 6: Visual comparison of the two datasets

images while not adversely affecting the diversity of the images. The image in row 4 demonstrates the sample with the best performance in terms of diversity and semantic consistency. The visual comparison results confirm the effectiveness of the DI and CE modules in improving the diversity and semantic consistency of image generation.

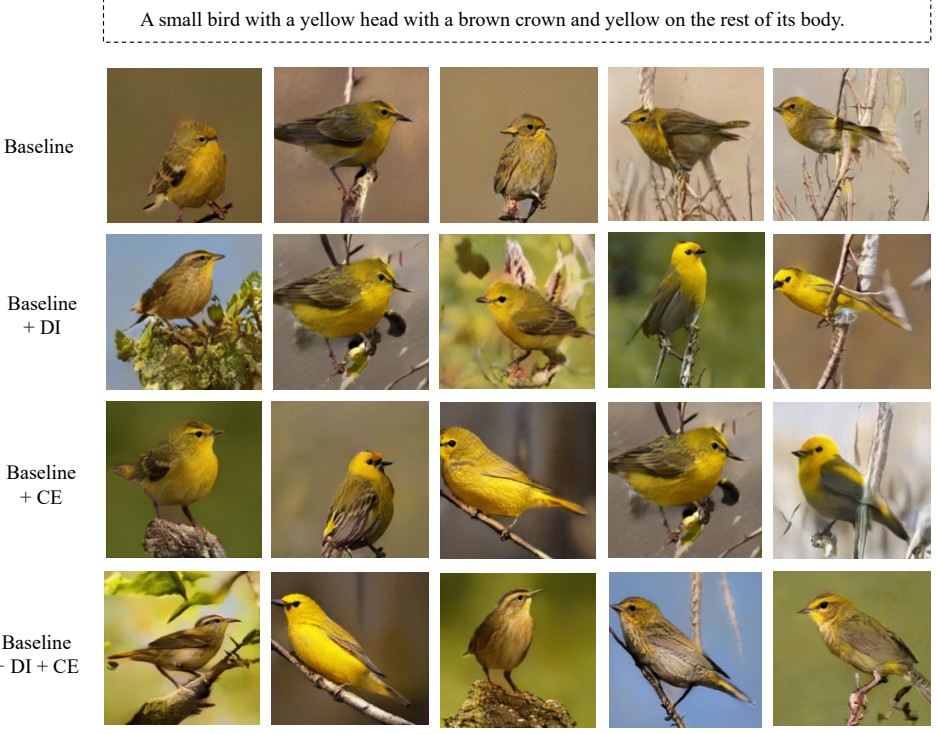

Figure 7: Visual comparison of ablation experiments

### A.0.3 IMPACT OF NOISE LEVEL ON PERFORMANCE

We find that the added noise broadcast can increase the stochastic diversity of the generated images, and the initial stage of the training uses noise vectors of lower dimensions and the dimensions of

the noise vectors are gradually increased with the training, and at the same time, to avoid the noise brings excessive randomness, we add a self-attention mechanism to the module to maintain the consistency of the global structure. Table 4 shows the performance test results of the model on the CUB dataset under varying noise intensities. We fixed the noise vector tuning intensity at 0.1 and assessed the IS, FID, and CLIPScore metrics of the resulting images. Specifically, at a noise intensity of $\sigma = 0.1$, the model exhibited enhanced performance metrics compared to the baseline. When $\sigma = 0.2$, there were incremental improvements: IS rose from 4.62 to 4.63, FID decreased from 19.40 to 19.02, and CLIPScore increased from 0.23 to 0.26. However, at $\sigma = 0.3$, excessive noise injection led to overly diverse images, diminishing their realism and causing a decline in the IS score. Additionally, heightened noise introduction at this level exacerbated discrepancies between the feature representations of generated and real images, thereby lowering image quality and deteriorating FID and CLIPScore metrics. Consequently, to strike a balance between image diversity and quality, we opted to set the noise vector intensity at 0.2 during the Dice-GAN model training phase.

Table 4: Effects of different noise intensities on performance

| $\sigma$ | IS↑ | FID↓ | CLIPScore↑ |
|---|---|---|---|
| 0.0 | 4.73 | 18.26 | 0.24 |
| 0.1 | 4.75 | 16.03 | 0.26 |
| 0.2 | 4.63 | 17.11 | 0.25 |
| 0.3 | 4.62 | 19.40 | 0.23 |

