# OpenReview forum: "Dice-GAN: Generative Adversarial Network  with Diversity Injection and Consistency Enhancement"
_ICLR.cc/2025/Conference — Submitted to ICLR 2025_

### Official Review · Reviewer_zzQP · 2024-10-16

**Soundness:** 2
**Presentation:** 2
**Contribution:** 1
**Rating:** 3
**Confidence:** 4

**Summary:**

he paper proposes Dice-GAN, an efficient attention-based text-to-image synthesis model. To enhance image diversity, a diversity injection module is introduced, incorporating noise and a self-attention mechanism. A consistency enhancement module, combining word vectors and a hybrid attention mechanism, improves semantic consistency. Experimental results on CUB and COCO datasets demonstrate Dice-GAN's superiority in image fidelity and diversity compared to existing approaches.

**Strengths:**

- Clear and well-organized presentation.
- Superior performance to other GAN-based methods.

**Weaknesses:**

- Limited novelty: While the diversity injection module is a contribution, the core idea of adding noise is not entirely novel.
- Lack of comparison to diffusion models: Given the current dominance of diffusion models in text-to-image generation, a more comprehensive comparison to state-of-the-art diffusion-based methods is essential to establish Dice-GAN's significance.
- Insufficient discussion of other generative models: The paper could benefit from a more in-depth discussion of how other generative models, such as flow-based models and StyleGAN, could be adapted or combined with Dice-GAN to further enhance diversity and quality.

**Questions:**

Please see weakness.

---

> ### Author Response · Authors · 2024-11-21
> **Response**
>
> Dear, reviewer, thank you very much for your valuable suggestions.
>
> According to the question you raised, our answer and modification are as follows. Please help us to see if this modification is OK.
>
> Q1. Limited novelty: While the diversity injection module is a contribution, the core idea of adding noise is not entirely novel.
>
> We acknowledge that adding noise to enhance diversity is not entirely new. However, we emphasize that the innovation of the DI module is to fuse the noise injection with the text information and balance the global structure through the self-attention mechanism, to promote diversity while maintaining the image quality. We will emphasize this more explicitly in the abstract. Specifically, our modification is as follows.
>
> **To improve the diversity of image generation, we design a diversity injection module, which injects noise several times during the image generation process, fuses the noise with the textual information, and incorporates a self-attention mechanism to help the generator maintain global structural consistency while enhancing the diversity of the generated image.**
>
> Q2. Lack of comparison to diffusion models: Given the current dominance of diffusion models in text-to-image generation, a more comprehensive comparison to state-of-the-art diffusion-based methods is essential to establish Dice-GAN's significance.
>
> We have supplemented the comparison experiments with recently proposed diffusion models such as CogView, DALL-E and ShiftDDPMs,  and conduct a detailed analysis in the experimental results section to fully demonstrate the advantages and applicable scenarios of Dice-GAN.
>
> Specifically, our modification of Section 4.2.1 is as follows.
>
> The comparison results of IS and FID across different models are shown in Table 1, where the best performance is shown in bold. By comparing with the stacked architectures AttnGAN, StackGAN, StackGAN++, StyleGAN models and the single-stage architectures DM-GAN, DF-GAN, and DE-GAN models on the CUB dataset, our Dice-GAN model demonstrates significant enhancement in the IS metrics from 4.28 to 4.93, and the FID metrics from 24.17 to 15.81, demonstrating that Dice-GAN exhibits excellent performance in IS and FID metrics. Converting to the MS-COCO dataset, Dice-GAN excels in FID performance, reducing it from 34.53 to 23.31. However, Dice-GAN slightly lags behind the other methods in terms of IS metrics. This discrepancy can be attributed to an inherent limitation of the IS metric (Zhang et al., 2021): the Inception model for IS computation was pre-trained on the ImageNet dataset, which is typically characterized for a single primary object, in contrast to the combinations of multiple objects that are often found in the MS-COCO dataset. This difference may lead to bias in IS assessment. We show the object differences between the ImageNet dataset and the COCO dataset in Figure 7 of the final appendix. Notably, our approach produces superior results to the diffusion models CogView, DALL-E, and ShiftDDPMs when compared to these models. In the above study, in addition to evaluating the image synthesis quality of the model, we also examined the operational efficiency of the model, especially the speed of image generation. The results show that the Dice-GAN model not only performs superiorly in terms of image quality and semantic consistency, but also significantly improves the efficiency of image generation. Specifically, compared to other models, Dice-GAN reduces the time for image generation from an average of 23.98 seconds to 9.02 seconds, which means that its image generation speed is improved by about 62.4%. This improvement significantly enhances the efficiency of image generation, making Dice-GAN more efficient and practical in practical applications.
>
> Q3. Insufficient discussion of other generative models: The paper could benefit from a more in-depth discussion of how other generative models, such as flow-based models and StyleGAN, could be adapted or combined with Dice-GAN to further enhance diversity and quality.
>
> We extend the discussion of stacked structure-based models and other generative models such as StyleGAN, StackGAN++, and StyleGAN in the Related work section. And we add new experiments to compared our approach with these methods.

---

> ### Author Response · Authors · 2024-11-25
> **Revision updated**
>
> Dear reviewer, the latest revised version is ready for your evaluation. Please help us to assess if the changes are satisfactory. If there are additional improvements we can make, please kindly let us know. Thank you again for the valuable suggestions.

---

> > ### Comment · Reviewer_zzQP · 2024-11-26
> >
> > How did you train CogView and DALL-E for the experiment? Additionally, it would be helpful to color-code the changes (e.g., using blue) to make it easier to identify the modified part in the revised version.

---

> > > ### Author Response · Authors · 2024-11-26
> > >
> > > Dear reviewer, we used the pre-trained versions of CogView and DALL-E. Given that the training process of these two models is extremely complex and requires significant computational resources, we obtained their pre-training weights from the open source community. We also include the description in the paper. According to your requirements, we have marked the modified part in blue, please download the new revision. Thank you!

---

> > > > ### Comment · Reviewer_zzQP · 2024-11-27
> > > >
> > > > Thank you for your response. However, it remains unclear whether a state-of-the-art T2I model trained on the same dataset as DiceGAN would outperform DiceGAN. I would recommend including such experiments as part of future work. I will keep my score unchanged.

---

> > > > > ### Author Response · Authors · 2024-11-27
> > > > >
> > > > > Dear reviewer, thank you for your suggestion. As far as we know, CogView and DALL-E trained their models on very large scale datasets, which are about millions of text-image pairs.  However, we trained our model on MS-COCO dataset and CUB dataset with the size of 400,000 and 80,000 text-image pairs, which are much smaller. So we believe that CogvVew and DALL-E have already got good enough parameters. Even if we trained CogView and DALL-E on our datasets, we believe the result could only be worse but not better, as the existing weights should already be sufficiently trained and optimized.

---

### Official Review · Reviewer_66gE · 2024-10-28

**Soundness:** 3
**Presentation:** 3
**Contribution:** 2
**Rating:** 6
**Confidence:** 5

**Summary:**

The manuscript introduces Dice-GAN which incorporates Diversity Injection and Consistency Enhancement modules to address critical challenges in generating high-quality, diverse images while maintaining semantic alignment with textual descriptions. Experimental results demonstrate that Dice-GAN outperforms state-of-the-art models on the CUB and MS-COCO datasets, underscoring its efficacy in enhancing visual quality and fidelity.

**Strengths:**

1. The introduction of the DI and CE modules marks a significant advancement in text-to-image synthesis. The DI module, which injects noise at multiple stages of generation, and the CE module, which integrates word vectors with hybrid attention, effectively improve both image diversity and semantic consistency.

2. This method achieves SOTA performance.

**Weaknesses:**

1. The manuscript lacks a detailed examination of the model's performance across varying levels of text complexity. Given that text descriptions can range from simple to highly nuanced, an analysis based on text complexity would provide stronger evidence of the model's robustness and its ability to handle diverse linguistic inputs.

2. The reviewer wants to see the experiment about computational efficiency.

3. The study does not thoroughly investigate the model's capacity to handle various textual attributes, such as color, size, and object positioning. A more focused evaluation of these specific attributes could offer deeper insights into the model's capability to accurately reflect detailed descriptive features and further demonstrate its adaptability.

**Questions:**

1. How does Dice-GAN perform under different levels of input noise? Given the pivotal role of the DI module, understanding the model's sensitivity to noise levels could provide valuable insights into balancing image diversity and visual quality effectively.

2. What measures were implemented to ensure that the DI module does not excessively degrade visual quality due to noise injection? A detailed discussion on the strategies used to balance noise injection and maintain visual quality would be beneficial.

3. Does the CE module exhibit limitations in maintaining semantic consistency for longer, more detailed text descriptions? An analysis of the CE module's performance with nuanced and complex descriptions would provide a clearer understanding of its efficacy in handling diverse linguistic inputs.

---

> ### Author Response · Authors · 2024-11-21
> **Response**
>
> Dear, reviewer, thank you very much for your valuable suggestions.
> According to the question you raised, our answer and modification are as follows. Please help us to see if this modification is OK.
>
> Q1. The manuscript lacks a detailed examination of the model's performance across varying levels of text complexity.
>
> We add the analysis of the image generation results with different text complexity in the experimental section, such as the comparison of simple and complex descriptions, to show the robustness of Dice-GAN when dealing with different language inputs.
>
> Q2. The reviewer wants to see the experiment about computational efficiency.
>
> We supplement the inference time experiments for image generation of the Dice-GAN model and compare with the contrast models to evaluate the practicality and efficiency of Dice-GAN.
>
> Q3. The study does not thoroughly investigate the model's capacity to handle various textual attributes, such as color, size, and object positioning.
>
> We conduct a more focused evaluation of specific text attributes such as color, size, and object localization, and analyze the ability of Dice-GAN to accurately reflect these descriptive features to more fully demonstrate the adaptability of Dice-GAN.
>
> Specifically, our modification of Section 4.2.3 is as follows.
>
> **In this case study, we analyzed the CUB and MS-COCO datasets and compared the outputs of AttnGAN, DMGAN, DF-GAN, DE-GAN, StackGAN, StyleGAN, and our proposed Dice-GAN, as shown in Figure 5. It is found that there are significant differences in the synthesis quality of these models. First, for the CUB dataset, the models have some problems in image generation under the long text of meticulous description. For example, the image in column 1 is missing body parts, the body shape is distorted in the image in column 2, the feather texture is confusing in column 3 and the incongruous body proportions in column 4. The differences in performance between models are still significant when using text with short descriptions, such as inconsistent colors in column 5, while column 6 results in greater diversity in the generated images due to the lack of specific descriptions of bird types, colors and sizes. In contrast, the Dice-GAN model demonstrates excellent image synthesis ability when processing both long and short text tasks. The model can effectively maintain the integrity and coherence of the subject in the image while generating details with natural gestures and realism, especially when generating bird images. In addition, when processing complex scene synthesis tasks from the MS-COCO dataset, Dice-GAN not only demonstrates excellent semantic consistency under long text descriptions, but can more accurately localize textual features, such as “Milkshake” in column 8 and “Train track” in column 10, but also demonstrates excellent semantic consistency under short text descriptions, such as “Milkshake” in column 8 and “Train track” in column 10. “It also ensures the harmony and accuracy of image feature localization in the face of short text descriptions, as demonstrated by “bedroom” in column 11 and “kite” in column 12. This shows that the Dice-GAN model is highly specialized in generating complex scenes with high fidelity.**
>
> Q4. How does Dice-GAN perform under different levels of input noise?
>
> We divide the ablation study into two parts to study the DI and CE modules respectively, so as to discuss the details in more depth. And we have added performance study under different levels of input noise as suggested in the ablation study.
>
> Q5. What measures were implemented to ensure that the DI module does not excessively degrade visual quality due to noise injection?
>
> We have added the discussion of strategies used to balance noise injection and maintain visual quality.
>
> **To avoid the noise brings excessive randomness, we add a self-attention mechanism to the module to maintain the consistency of the global structure. In addition, the experiments also show that too many feature fusion layers will increase the computational burden and the effect is not good, so we finally choose to add two feature fusion layers.**
>
> Q6. Does the CE module exhibit limitations in maintaining semantic consistency for longer, more detailed text descriptions?
>
> We have added discussion in the ablation experiment.
>
> **In the ablation experiments, the addition of either the conditional channel attention mechanism or the spatial attention mechanism alone ignored some of the information to some extent, and the consistency of the model was significantly enhanced after combining the two mechanisms in the CE module for the experiments. In addition, the experiments found that the resolution of the image features generated in the early stage of the model generation is small, the effect of adding the CE module on the consistency enhancement is not obvious, and it will increase the computation time of the model.**

---

> > ### Comment · Reviewer_66gE · 2024-11-23
> > **Response**
> >
> > The reviewer appreciates the author's response, although I think more experiments speak louder than words. I raise my score to 6 and encourage the author to provide more experiments.

---

> > > ### Author Response · Authors · 2024-11-25
> > > **Revision updated**
> > >
> > > Dear reviewer, thank you so much. We have prepared the latest revised version. Please help us to assess if the changes are satisfactory. If there are additional improvements we can make, please kindly let us know. Thank you again for the valuable suggestions.

---

### Official Review · Reviewer_TDGk · 2024-10-30

**Soundness:** 2
**Presentation:** 2
**Contribution:** 2
**Rating:** 3
**Confidence:** 3

**Summary:**

In this work, they propose the diversity injection and consistency enhancement module for text-to-image generation. This method contribute to produce high-quality images with increased diversity and enhanced semantic consistency based on text descriptions.

**Strengths:**

1. Enhanced Diversity: The Diversity Injection module injects noise and text vectors multiple times, ensuring a broad range of image outputs without sacrificing structure.

2. Improved Consistency: The Consistency Enhancement module dynamically adjusts focus on image regions, aligning visuals closely with text descriptions.

**Weaknesses:**

1. A comparison with recently proposed text-to-image generation models is needed. Not only should there be an analysis of issues with GANs, but also recent Diffusion models, along with performance comparisons. Is there a specific reason you only compared with ShiftDDPMs in the case of Diffusion models? Please provide a detailed response.

2. Please provide a detailed explanation of the table and figure captions.

3. Performance comparisons on diverse datasets are required. Additionally, besides IS and FID, comparisons with other performance metrics are requested (e.g., CLIP score).

4. The examples of qualitative results are too limited.

5. There is a lack of experimental analysis demonstrating the effectiveness of the proposed model structure.

**Questions:**

Please, see the weakness.

---

> ### Author Response · Authors · 2024-11-21
> **Response**
>
> Dear, reviewer, thank you very much for your valuable suggestions.
>
> Based on the question you raised, our answer and modification are as follows: Please help us to check if this modification is OK.
>
> Q1. A comparison with recently proposed text-to-image generation models is needed. Not only should there be an analysis of issues with GANs, but also recent Diffusion models, along with performance comparisons. Is there a specific reason you only compared with ShiftDDPMs in the case of Diffusion models? Please provide a detailed response.
>
> We have supplemented the comparison experiments with recently proposed diffusion models such as CogView, DALL-E and ShiftDDPMs, and conducted a detailed analysis in the experimental results section to fully demonstrate the advantages and applicable scenarios of Dice-GAN.
>
> Q2. Please provide a detailed explanation of the table and figure captions.
>
> We have carefully checked the table and figure titles to make sure they are clear and easy to understand and provide necessary explanations and instructions.
>
> Q3. Performance comparisons on diverse datasets are required. Additionally, besides IS and FID, comparisons with other performance metrics are requested (e.g., CLIP score).
>
> We have added performance comparisons in Section 4.2.1. In addition to IS and FID, we supplement the performance metric LPIPS value to more comprehensively evaluate the diversity of the generated results of Dice-GAN.
>
> Specifically, our modification is as follows.
>
> **To fully evaluate the performance of the DI module in improving image diversity, we computed the average LPIPS distance between 3K pairs of images, each generated from the same sentence. Higher LPIPS values indicate greater differences between images, thus reflecting better diversity. The results of the ablation experiments on the CUB dataset are shown in Table 2.**
>
> **Impact of DI: The integration of the DI module significantly improves the image generation quality of the model, with an IS value of 4.81 and a FID value of 18.37. Through the ablation experiments, we find that the added noise broadcast can increase the stochastic diversity of the generated images, and the initial stage of the training uses noise vectors of lower dimensions and the dimensions of the noise vectors are gradually increased with the training, and at the same time, to avoid the noise brings excessive randomness, we add a self-attention mechanism to the module to maintain the consistency of the global structure. In addition, the experiments also show that too many feature fusion layers will increase the computational burden and the effect is not good, so we finally choose to add two feature fusion layers to enhance the fusion effect of text information and image features while maintaining the computational efficiency. Together, these optimization measures improve the ability of the DI module to generate high-fidelity and diverse images, further proving the effectiveness and practicality of the DI module.**
>
> **Effect of CE: After combining the CE module, the IS value increased from 4.62 to 4.65, and the FID value decreased significantly from 19.40 to 16.24. In the ablation experiments, the addition of either the conditional channel attention mechanism or the spatial attention mechanism alone ignored some of the information to some extent, and the consistency of the model was significantly enhanced after combining the two mechanisms in the CE module for the experiments. In addition, the experiments found that the resolution of the image features generated in the early stage of the model generation is small, the effect of adding the CE module on the consistency enhancement is not obvious, and it will increase the computation time of the model. Therefore, we chose to add the CE module at the stage with a resolution of 64 × 64, which can significantly improve the semantic consistency of the model-generated images while maintaining computational efficiency. These findings confirm the effectiveness and usefulness of the proposed CE module in improving the semantic consistency of model-generated images.**
>
> Q4. The examples of qualitative results are too limited.
>
> We have added more examples of qualitative results to show the performance benefits of Dice-GAN more intuitively.
>
> Q5. There is a lack of experimental analysis demonstrating the effectiveness of the proposed model structure.
>
> We have supplemented this with a more detailed experimental analysis to verify the effectiveness of the DI and CE modules and explain their impact on the model performance. We divide the ablation study into two parts, studying the DI and CE modules respectively, so as to discuss the details in more depth.

---

> > ### Comment · Reviewer_TDGk · 2024-11-25
> >
> > We appreciate the author's response and have increased the score. However, I believe the paper still lacks a thorough analysis of DiceGAN and comprehensive comparisons with state-of-the-art diffusion models.

---

> > > ### Author Response · Authors · 2024-11-25
> > > **Revision updated**
> > >
> > > Dear reviewer, Thank you so much. We have prepared the latest revised version. In addition to the changes mentioned before, we also added the CLIPScore evaluation metric as you suggested in the new revision.  Please help us to assess if the changes are satisfactory. If there are additional improvements we can make, please kindly let us know. Thank you again for the valuable suggestions.

---

### Official Review · Reviewer_r6db · 2024-11-04

**Soundness:** 3
**Presentation:** 2
**Contribution:** 2
**Rating:** 5
**Confidence:** 4

**Summary:**

This work proposes DICE-GAN, a single-stage text-to-image GAN to produce high-quality and high-diversity images with improved semantic consistency with text condition. The paper proposes two modules: The Diversity Injection (DI) module, which adds learnable noise to the image features for increasing diversity in generated images, and the Consistency Enhancement (CE) module, which allows the model to dynamically adjust the weights of different image features according to input text conditions for improved semantic consistency and fidelity.

--
The authors have provided the ablation study for the DI module on the CUB dataset with a small improvement on the IS metric. However, it is unclear if these gains will be present when scaling to larger datasets like COCO or Imagenet.
The presented argument for the novelty of the DI module is not new: "injects noise several times during the image generation process, fuses the noise with the textual information, and incorporates a self-attention mechanism to help the generator maintain global structural consistency." Further, the reported IS score is low compared to AttnGAN and DM-GAN on the MS-COCO dataset and is missing a full-scale comparison with Imagenet.

**Strengths:**

1. The idea of adding learnable noise in different training phases and correction with self-attention to improve generation diversity is novel and interesting.

2. The authors demonstrate improved performance on the IS and FID metrics on the CUB dataset and on the FID metric on the MS-COCO dataset.

3. The authors provide an ablation study demonstrating improvements in results by adding Diversity Injection (DI) and Consistency Enhancement (CE) modules.

**Weaknesses:**

1. The novelty of the work is limited. The idea of feature fusion in Eq 1 in the DI module is not novel and has been explored before[1,2,3] in the context of image generation. Further, the idea of masking features in a condition-dependant manner has limited novelty. 2. Lack of clarity in Sec 3.2 writing and Fig 4. The idea behind Conditional Channel Attention mask($M_c$) and Spatial Attention attention($M_s$) is unclear. The motivation behind generating masks from both average and max channels is also unclear. Further, quantities including $G^{c}_{max}$ and $G^{c}_{avg}$ are missing in Fig 4, making it difficult to understand figure pipeline. 3. The authors claim that Dice-GAN utilizes a single-stage model structure for improved performance but are missing comparisons with multi-stage methods, including StackGAN++[4]. 4. Missing ablation studies: - Why are two feature fusion layers are needed in the DI module? How was this hyperparameter determined? - How does learnable noise $\sigma$ vary when going from lower to higher layers in the trained model? - Missing ablation on design choices in CE module on use of average and max features and conditional channel attention and spatial attention submodule. 5. The proposed method achieves a lower IS score on the MS-COCO dataset, and the authors argue that this is due to the Inception model used in IS computation being pre-trained on the ImageNet dataset. The authors should provide results on Imagenet or Imagenet subset to back their claims.

[1] Ethan Perez, Florian Strub, Harm De Vries, Vincent Dumoulin, and Aaron Courville. Film: Visual reasoning with a general conditioning layer. In AAAI, 2018. 2, 5
[2] Tero Karras, Samuli Laine, and Timo Aila. A style-based generator architecture for generative adversarial networks. In CVPR, 2019. 5
[3] Peebles, William, and Saining Xie. "Scalable diffusion models with transformers." Proceedings of the IEEE/CVF International Conference on Computer Vision. 2023.
[4] Zhang, Han, et al. "Stackgan++: Realistic image synthesis with stacked generative adversarial networks." IEEE transactions on pattern analysis and machine intelligence 41.8 (2018): 1947-1962.

**Questions:**

Please see weaknesses

---

> ### Author Response · Authors · 2024-11-21
> **Response**
>
> Dear, reviewer, thank you very much for your valuable suggestions.
>
> According to the question you raised, our answer and modification are as follows. Please help us to see if this modification is OK
>
> Q1. The novelty of the work is limited.
>
> We acknowledge that adding noise to enhance diversity is not entirely new. However, we emphasize that the innovation of the DI module is to fuse the noise injection with the text information and balance the global structure through the self-attention mechanism, to promote diversity while maintaining the image quality. We will emphasize this more explicitly in the abstract. Specifically, our modification is as follows.
>
> **To improve the diversity of image generation, we design a diversity injection module, which injects noise several times during the image generation process, fuses the noise with the textual information, and incorporates a self-attention mechanism to help the generator maintain global structural consistency while enhancing the diversity of the generated image.**
>
> Q2. Lack of clarity in Sec 3.2 writing and Fig 4.
>
> We have improved the writing of Section 3.2 and provide a clearer Figure 4 for a better understanding of how the CE module works. Specifically, our modification is as follows.
>
> **To enhance the consistent generation of image features and textual information, we consider improving the model from both channel and spatial perspectives. In the Consistency Enhancement (CE) module, we successfully integrate the word vector $W$ into the conditional channel attention mechanism, which is used to identify and enhance the most important feature channels in the generator to improve the quality of the generated images. By learning the importance of each channel, the model can pay more attention to the information that is crucial for image generation while suppressing irrelevant features. This is combined with a spatial attention mechanism to ensure that high-level and low-level features complement each other in generating the image, enhancing the detail and structure of the image. This integration aims to improve the visual quality throughout the image generation process. Figure 4 provides a visual representation of the integrated structure of the CE module.**
>
> In the paragraph of Hybrid attention feature generation stage. We add following sentences.
>
> **The maximum pooling operation retains the maximum values for each channel, which represent the most salient features in the feature map, such as critical parts of the image or edge information. The average pooling operation calculates the average of all values for each channel, which reflects the overall characteristics of the feature map. Average pooling captures the global information in the feature map, including background and texture. Thus, it can preserve the background information in the feature map and help the model better understand the overall structure.**
>
> In the second paragraph of Hybrid attention feature generation stage. We will add following sentence before “The computational steps are outlined in Equation 3 and Equation 4.”
>
> **Channel attention weights $M_c$ is generated to consolidate information across the complete feature map. This process enhances the significance of crucial channels while diminishing the influence of less critical ones, thereby enhancing model efficiency. In $M_c$, each element signifies the weight of the respective channel in the feature map, derived by summing the average pooling weight and the maximum pooling weight of that specific channel. A greater weight value denotes increased channel importance in message conveyance.**
>
> In the third paragraph, we will add following sentence after “Subsequently, a sigmoid function is applied to the output to obtain the spatial attention map Ms.”
>
> **$M_s$ is employed to pinpoint spatial positions within the feature map, highlighting key local regions essential for image synthesis. This approach enables the model to concentrate on intricate details and textures during image generation, leveraging contextual cues to produce more nuanced and contextually rich images by directing its focus toward distinct image regions.**
>
> Q3. Missing comparisons with multi−stage methods, includingStackGAN++[4].
>
> We have supplemented the comparison experiments with multi-stage methods such as StackGAN, StackGAN++ and StyleGAN.
>
> Q4. Missing ablation studies
>
> We have supplemented ablation experiments, such as analyzing the impact of the number of feature fusion layers in the DI module and different submodules in the CE module, to verify the effectiveness of the model design.
>
> Q5. Provide results on Imagenet or Imagenet subset to back their claims.
>
> We have added a Figure 7 to show a comparison of images on the ImageNet dataset on which the Inception model has been pre-trained and on MS-COCO, where the ImageNet dataset has features for a single primary object, as opposed to the MS-COCO dataset, which is often a combination of multiple objects.

---

> ### Author Response · Authors · 2024-11-25
> **Revision updated**
>
> Dear reviewer, the latest revised version is ready for your evaluation. Please help us to assess if the changes are satisfactory. If there are additional improvements we can make, please kindly let us know. Thank you again for the valuable suggestions.

---

### Meta-Review · Area_Chair_oDrJ · 2024-12-17

**Metareview:**

The paper introduces Dice-GAN, a novel text-to-image generation model that incorporates a Diversity Injection (DI) module to enhance image diversity and a Consistency Enhancement (CE) module to improve semantic alignment. Experimental results on CUB and MS-COCO datasets demonstrate that Dice-GAN outperforms state-of-the-art models in visual quality and fidelity. However, the novelty of this paper is marginal (e.g., the DI module's feature fusion approach, masking features). Besides, this paper lacks sufficient comparisons with multi-stage methods and state-of-the-art diffusion models (CogView and DALL-E) on different datasets or cross datasets to show the generalization ability of the proposed method. The rebuttal did not fully address the reviewers' problems, and all the reviewers lean towards rejecting this paper.

**Additional Comments On Reviewer Discussion:**

Reviewers raised concerns about the limited novelty of the Diversity Injection module, lack of comparisons to diffusion models, incomplete ablations, and unclear figures. The authors clarified design choices and partially addressed ablation and clarity issues but did not provide sufficient new evidence or comparisons. While the paper demonstrates merit in improving image diversity and semantic consistency, the unresolved novelty concerns and missing comprehensive experiments weighed heavily in the final decision.

---

### Decision · Program_Chairs · 2025-01-22

Reject